# TNFα Rescues Dendritic Cell Development in Hematopoietic Stem and Progenitor Cells Lacking C/EBPα

**DOI:** 10.3390/cells9051223

**Published:** 2020-05-15

**Authors:** Subramanian Anirudh, Angelika Rosenberger, Elke Schwarzenberger, Carolin Schaefer, Herbert Strobl, Armin Zebisch, Heinz Sill, Albert Wölfler

**Affiliations:** 1Division of Hematology, Medical University of Graz, Auenbruggerplatz 38, A-8036 Graz, Austria; anirudh.muralikrishnan@medunigraz.at (S.A.); angelika.rosenberger@medunigraz.at (A.R.); carolin.schaefer@medunigraz.at (C.S.); armin.zebisch@medunigraz.at (A.Z.); heinz.sill@medunigraz.at (H.S.); 2Division of Immunology and Pathophysiology, Otto Loewi Research Center for Vascular Biology, Immunology and Inflammation, Medical University of Graz, Heinrichstraße 31, A-8010 Graz, Austria; el.baumgartner@medunigraz.at (E.S.); herbert.strobl@medunigraz.at (H.S.); 3Division of Pharmacology, Otto-Loewi-Research Center for Vascular Biology, Immunology and Inflammation, Medical University of Graz, Universitätsplatz 4, A-8010 Graz, Austria

**Keywords:** dendritic cell development, monocytic dendritic cell progenitors, C/EBPα, TNFα

## Abstract

Dendritic cells (DCs) are crucial effectors of the immune system, which are formed from hematopoietic stem and progenitor cells (HSPCs) by a multistep process regulated by cytokines and distinct transcriptional mechanisms. C/EBPα is an important myeloid transcription factor, but its role in DC formation is not well defined. Using a *Cebpa*^Cre^-EYFP reporter mouse model, we show that the majority of splenic conventional DCs are derived from *Cebpa*-expressing HSPCs. Furthermore, HSPCs isolated from *Cebpa* knockout (KO) mice exhibited a marked reduced ability to form mature DCs after in vitro culture with FLT3L. Differentiation analysis revealed that C/EBPα was needed for the formation of monocytic dendritic progenitors and their transition to common dendritic progenitors. Gene expression analysis and cytokine profiling of culture supernatants showed significant downregulation of inflammatory cytokines, including TNFα and IL-1β as well as distinct chemokines in KO HSPCs. In addition, TNFα-induced genes were among the most dysregulated genes in KO HSPCs. Intriguingly, supplementation of in vitro cultures with TNFα at least partially rescued DC formation of KO HSPCs, resulting in fully functional, mature DCs. In conclusion, these results reveal an important role of C/EBPα in early DC development, which in part can be substituted by the inflammatory cytokine TNFα.

## 1. Introduction

Dendritic cells (DCs) are sentinels of the immune system that function via the uptake of antigens and presenting antigenic peptides to naïve T-cells, leading to T-cell priming and effector cell differentiation [1]. DCs are classified based on origin, location, and function into steady state DCs, which encompass conventional DCs (cDCs) and plasmacytoid DCs (pDCs), as well as Langerhans cells (LCs) and monocyte-derived DCs. Apart from LCs, all types of DCs arise from bone marrow (BM)-resident hematopoietic stem and progenitor cells (HSPCs), which undergo sequential differentiation steps via lineage-restricted progenitors. According to the current model, steady-state DC development starts with the formation of monocytic dendritic cell progenitors (MDPs) from multipotent (MPPs) or common myeloid progenitors (CMPs) [2,3,4,5,6] through the activity of the cytokine FMS-related tyrosine kinase-3 ligand (FLT3L). MDPs then give rise to common dendritic cell progenitors (CDPs). The CDPs eventually differentiate into all types of cDCs and pDCs [3]. The stepwise differentiation of progenitors is regulated by intrinsic transcription factor (TF) networks as well as extrinsic mechanisms controlled by cytokines. For example, the ETS factor PU.1 acts as a master regulator during the transition of MPPs to MDPs and is induced by FLT3L [7,8]. Moreover, TFs such as IRF8, IRF4, and ID2 [9,10] have been shown to have effects on multiple steps of DC development.

Another important TF in hematopoiesis is CCAAT/enhancer binding protein alpha (C/EBPα). While its C-terminal basic region leucine zipper (BR-LZ) domain mediates DNA-binding and protein–protein interactions with other transcription factors, the two transactivation domains are responsible for E2F repression and represent binding sites for proteins such as SWI/SNF and CDK2/CDK4 [11]. C/EBPα plays a well-defined role in the formation of myeloid progenitors [12,13,14] and elicits a lineage-instructive function in MPPs towards the myeloid lineage [15]. Although C/EBPα has been identified to play a role in early but not late stages of DC differentiation [16], little is known about its mechanisms in DC development. Here, we show that the majority of cDCs in the spleen are indeed derived from *Cebpa*-expressing HSPCs and that C/EBPα is needed for the formation of MDPs and their transition to CDPs. We identify an important interplay between C/EBPα and TNFα in early DC development: HSPCs lacking *Cebpa* expression displayed a prominent dysregulation of TNFα-induced genes, were blocked in DC formation in vitro, and could be rescued by addition of TNFα.

## 2. Materials and Methods

### 2.1. Mice

*Mx1^Cre^/Cebpa^F/F^* mice were obtained from Prof. Daniel G Tenen, Department of Hematology, Harvard University, Boston, MA, USA. As described [13,16], treatment of these mice (aged between 8–10 weeks) with four injections of polyinosinic:polycytidylic acid (pIpC) every two days results in full bone marrow-specific knock-out (KO) of *Cebpa* (see Appendix A). Mice were analyzed 2–3 weeks after the last injection. The *Cebpa^Cre^-*EYFP (enhanced yellow fluorescence protein) reporter mouse model was used to trace *Cebpa*-expressing cells and their progeny in various hematopoietic cell compartments as described [15]. All mice were on a C57Bl/6 background and housed in specific pathogen-reduced conditions at the central animal facility of the Medical University of Graz. Mouse experiments were approved by the Austrian Federal Ministry for Science, Research, and Economy (GZ: BMWF-66.010/0017-II/3b/2014).

### 2.2. Primary HSPC Cultures and Cytokine Treatment

Bone marrow (BM) cells were isolated from both *Cebpa* wildtype (*Cebpa^F/F^*) and *Cebpa* knockout (*Mx1^Cre^/Cebpa^F/F^*) mice (hereafter referred to as WT and KO, respectively) as described [15,17]. Fibrous material was removed by filtration through a 70-μm nylon mesh and erythrocytes were lysed using PharmLyse buffer (BD Biosciences, San Jose, CA, USA). Lineage negative (lin^−^) HSPCs were then isolated by depletion of BM cells expressing the lineage markers Ter119, CD11b, Gr-1, CD11c, CD3e, and B220 using the IMag Hematopoetic Progenitor Enrichment Kit (BD Biosciences) by magnetic separation according to the manufacturer’s instructions. For in vitro DC development, HSPCs (10^6^/mL) were cultured in 24 well plates in RPMI1640 medium supplemented with 10% fetal calf serum, 1% PenStrep, 0.1% B-ME, and FLT3L (200 ng/mL) for 8 days [18]. Based on the experimental set-up, some cultures were supplemented with either TNFα, MIP-1α, or MIP-2 (all 20 ng/mL). All cytokines and chemokines were purchased from Peprotech (Rocky Hill, NJ, USA). Experiments were repeated independently at least once, if not otherwise stated, and number of mice is given in the figure legends.

### 2.3. Flow Cytometry Analysis and Cell Sorting

Flow cytometric analysis was done as described [15]. Briefly, samples were analyzed as single cell suspensions with different antibody panels (for information on used antibody clone and fluorochromes, see Appendix A). To analyze EYFP expression in various progenitor compartments, cells were stained with the following lineage markers: CD3e, CD11b, CD11c, CD45R/B220, Gr1, and Ter119. CMPs were defined as Lin^–^CD117^+^CD34^+^CD16/32^−^ and GMPs as Lin^−^CD117^+^CD34^+^CD16/32^+^. MPPs were defined as Lin^−^FLT3^+^CD117^+^CD115^−^, MDPs as Lin^−^FLT3^+^CD117^+^CD115^+^, and CDPs as Lin^−^FLT3^+^CD117^lo/int^CD115^+^. In vitro FLT3L-induced mature DCs were defined as CD11c^+^MHCII^+^ cells (for gating see [15] and Appendix A). For analysis of DC progenitors during FLT3L stimulation in vitro, staining for FLT3 was not possible due to FLT3L-induced FLT3-internalization. Cells were analyzed using a LSRII flow cytometer (BD Biosciences), and sorting was performed on a FACS Aria II (BD Biosciences). Purity of double-sorted population was >98%. Flow cytometry data analysis was done using Kaluza software (Beckman Coulter, Krefeld, Germany).

### 2.4. Gene Expression Analysis and Quantitative Real Time (RT)-PCR

Double sorted lin^-^CD117^+^FLT3^+^ HSPCs from 4 WT and KO mice were used for gene expression analysis. The cells were treated (T) with FLT3L or left untreated (UT) for 4 h. RNA was isolated from the WT/KO cells with or without FLT3L treatment (referred to hereafter as WT(T)/WT(UT) and KO(T)/KO(UT)) using the RNeasy Micro Kit (Qiagen, Hilden, Germany). The mRNA amplification and hybridization was done using the Affymetrix Mouse Gene 1.0 ST array as per manufacturer instructions (Affymetrix, Santa Clara, CA, USA). The hybridization and expression data were analyzed on Gene-expression Console v.1.1, Affymetrix. The gene expression data were further analyzed in detail using BioVenn [19] and Network Analyst web-based freeware [20,21]. All microarray expression data are available at GEO: GSE146288. Results from the microarray were confirmed using qRT-PCR. RNA was isolated from cultured HSPCs using RNeasy Micro Kit (Qiagen) according to the manufacturer’s protocol. cDNA synthesis was done with 200 ng of RNA using the High Capacity cDNA kit (Applied Biosystems, Foster City, CA, USA). The cDNAs were then assayed with TaqMan (Applied Biosystems) probes for *Cebpa*, *Cx3cr1*, *B-Actin*, and *Gapdh* (Appendix A). The expression was calculated using the ΔΔCt method and represented as x-fold change.

### 2.5. Cytokine Profiling Assay

Lin^−^ HSPCs were isolated from both WT and KO mice (*n* = 6 mice per group). The cells were then set in culture for 6 h with 200 ng/mL of FLT3L. After 6 h, the supernatants from different cultured cells were collected for multiplex cytokine analysis. One hundred microliters of each supernatant was used to detect the difference in the cytokine profile in a 96-well plate format using the Bio-Plex™ system (Biorad Laboratories Inc., Hercules, CA, USA) with a custom cytokine panel (Appendix A).

### 2.6. Allogenic Mixed Lymphocyte Reaction (MLR) Assay

MLR assays were done as described [22]. In brief, in vitro generated DCs from WT and KO HSPCs were mixed with splenic T-cells from BALB/c mice. 10^5^ T-cells were seeded with increasing numbers of DCs (10^3^, 3 × 10^3^, 10^4^, and 3 × 10^4^) in triplicates. After 5 days of co-culture, radioactive thymidine was added to the culture and uptake was calculated as a measure of T-cell proliferation after 16 h.

### 2.7. Statistical Analysis

Prism 6 software (GraphPad, La Jolla, CA, USA) was used to assess the statistical differences between two groups with a two-sample t-test with Welch’s correction (two-tailed). All results are presented as the mean ± SD, and *p* < 0.05 was considered statistically significant.

## 3. Results

### 3.1. C/EBPα Is Expressed in Early DC Progenitors and Indispensable for FLT3L-Induced DC Formation

To trace *Cebpa* expression in DCs and its progenitors in vivo, we used the *Cebpa*^Cre^-EYFP reporter mouse model, which enables tracking of cells that either express *Cebpa* or are the progeny of *Cebpa*-expressing (*Cebpa*/EYFP^+^) cells in steady-state hematopoiesis [15]. With this approach, we detected *Cebpa*/EYFP expression in about 60% of mature CD11c^+^MHCII^+^ cDC in the spleen (Figure 1). Since *Cebpa* is hardly expressed in mature DCs [16], these results suggest that the majority of spleen cDCs is derived from *Cebpa*/EYFP expressing progenitors. Indeed, in the bone marrow we observed an increasing percentage of *Cebpa*/EYFP^+^ cells starting from MPPs via CMPs to MDPs (Figure 1) similar to what has been reported during granulopoiesis towards GMPs [15]. Interestingly, there was no additional increase in *Cebpa*/EYFP^+^ cells after the stage of MDPs, which is in accordance with the finding that C/EBPα is completely dispensable for murine DC development at later stages [16]. Furthermore, the percentage of *Cebpa*/EYFP^+^ cells corresponded well to reported actual *Cebpa* mRNA expression levels in progenitor populations [23] (see Appendix A).

Given the fact that the majority of MDPs were positive for *Cebpa*/EYFP expression, we wondered whether C/EBPα plays an important role in (early) DC development. For this purpose, we studied in vitro DC development of primary bone marrow HSPCs isolated from inducible *Cebpa* knock-out (KO) mice (*Mx1^Cre^/Cebpa^F/F^*) and their wildtype (WT) littermates (*Cebpa^F/F^*) after FLT3L stimulation [18]. While HSPCs isolated from WT mice displayed robust formation of mature DCs as detected by CD11c and MHCII-expression after 8 days of culture with FLT3L [18], HSPCs isolated from *Cebpa* KO mice showed a significant reduction of almost 80% of their potential to form mature DCs (Figure 2A). These results indicate that C/EBPα is indeed needed in FLT3L-induced DC formation in vitro. Next, we performed a stepwise analysis of the different DC progenitor stages using FLT3L-stimulated in vitro cultures of HSPCs. Although we observed higher numbers of CD117^+^FLT3^+^ progenitors in *Cebpa* KO mice, which is consistent with the finding that lack of *Cebpa* results in increased formation of FLT3^+^ MPPs [24], the percentage of cells undergoing maturation towards CD117^hi^CD115^+^ MDPs was reduced and almost no CD117^lo/int^CD115^+^ CDPs were formed on days 1 and 3 in cultures with *Cebpa* KO HSPCs (Figure 2B). These results indicate that decreased DC formation of HSPCs lacking C/EBPα after FLT3L-stimulation is most likely due to reduced formation of MDPs and a block in transition to CDPs. However, since C/EBPα induces CD115 expression, the marker defining MDPs and CDPs, we cannot rule out that the identification of MDPs and CDPs is hampered in the *Cebpa* KO setting.

### 3.2. Lack of C/EBPα Alters the Cytokine/Chemokine Expression and Secretion Landscape in Early DC Progenitors

To identify pathways and target genes involved in C/EBPα-dependent mechanisms in early DC development, we next performed gene expression analysis of sorted FLT3^+^CD117^+^ HSPCs of WT and KO mice either untreated or stimulated with FLT3L for 6 h. Although the number of differentially expressed genes (>1.5 fold change, *p* < 0.05 after FDR correction) after FLT3L stimulation was similar in WT and KO FLT3^+^ HSPCs (2149 vs. 2128 genes, respectively), only 1104 genes overlapped, suggesting a profound role of C/EBPα in FLT3L-induced DC formation (Appendix A). To further understand the biological effects of *Cebpa* deletion, we did a comprehensive pathway analysis of differentially expressed genes. To get a full picture, we analyzed a list of genes that contained genes that showed difference exclusively in the presence of *Cebpa* (WT vs. WT (T) setting (1045)) and genes from the overlapping list that showed differential expression between the WT and KO setting (230/1104, see also Appendix A). Interestingly, these genes were linked to the TNFα signaling cascade, the NF-κB signaling pathway, cytokine–cytokine receptor interactions, and hematopoietic cell lineage developmental pathways (Figure 3). Further analysis of the exclusive WT vs. WT (T) (1045 genes) and the exclusive KO vs. KO (T) (1024 genes) lists (Appendix A) revealed that genes from the WT vs. WT (T) list were again found to be linked to the TNFα signaling cascade and the NF-κB signaling pathway, which were both absent in the KO vs. KO (T) gene list. These results indicate that lack of *Cebpa* significantly affects TNFα- and NF-κB signaling pathways in DC development. Interestingly, besides genes linked to these signaling pathways, TNFα itself and other cytokines and chemokines, like IL-1β, CCL5, and CCL3, were affected by the lack of *Cebpa* (Appendix A). Interestingly, when we performed pathway analysis comparing unstimulated WT and KO HSPCs, the TNFα-signaling pathway was already significantly deregulated (see Appendix A), suggesting hampered TNFα-signaling in general in KO HSPCs, which gets more pronounced after FLT3L-stimulation.

Having observed an effect on the transcriptional level of multiple cytokines and cytokine-related pathways, we questioned whether a change can be seen in the secretory profiles of HSPCs from WT and KO mice. For this purpose, we analyzed the supernatants of WT and KO HSPCs treated with FLT3L for the presence of various chemokines and cytokines. We observed a stark reduction of proinflammatory chemokines like CCL3 (MIP-1α), CXCL2 (MIP-2), CCL4 (MIP-1β), and CCL5 (RANTES) in the KO setting (Figure 4). Although not as pronounced as with the mentioned chemokines, inflammatory cytokines, like TNFα and IL-1β, were also significantly reduced in supernatants of KO, as compared to WT HSPCs. Interestingly, cytokines like CSF2 (GM-CSF) and CSF1 (M-CSF) showed no C/EBPα-dependent changes (Figure 4).

### 3.3. TNFα Rescues FLT3L-Induced DC Development in the Absence of Cebpa

From these results, we hypothesized that addition of these chemokines/cytokines to FLT3L stimulated cultures could rescue DC development in KO HSPCs. While treatment with MIP-1α (CCL3) or MIP-2 (CXCL2, Appendix A) had no effect on the potential of KO HSPCs to form DCs, a combination of FLT3L and TNFα induced enhanced development of KO towards mature CD11c^+^MHCII^+^ DCs, as compared to the KO HSPCs treated only with FLT3L (Figure 5, Appendix A). We observed an almost three-fold increase of mature DCs formed with the combinatorial treatment, compared to with FLT3L alone, which eventually resulted in comparable numbers of DCs as obtained from WT HSPCs treated with FLT3L. Interestingly, DCs formed in the presence of TNFα showed higher CD11c and MHCII expression, as described [25]. To prove that the DCs generated with this combination treatment from KO HSPCs were mature DCs, we checked for cellular morphology and found them to be comparable to the DCs generated from WT HSPCs (Figure 6A). We also performed an allogenic mixed lymphocyte reaction to check for their functionality and observed that DCs generated by treating KO HSPCs with FLT3L and TNFα could indeed activate naïve T-cells in a comparable manner to WT cells obtained with FLT3L (Figure 6B). In contrast, we did not get sufficient numbers of mature DC from KO HSPCs treated with FLT3L alone to perform the assay.

Since expression of *Cx3cr1* is among the first characteristics attributable to MDPs and we have observed a prominent downregulation of this gene in KO, as compared to WT HSPCs, we wondered whether addition of TNFα to FLT3L could restore *Cx3cr1* expression in KO HSPCs. Indeed, combination treatment including both cytokines increased *Cx3cr1* expression in KO HSPCs (Figure 6C). This suggests that additional TNFα stimulation might indeed bypass the early differentiation block in MDPs caused by the lack of *Cebpa*.

## 4. Discussion

Development of steady-state DCs is a complex process, starting with BM-resident HSPCs and involving the action of several cytokines and distinct transcription factors [26,27]. While some cytokines and their receptors, such as GM-SCF (CSF2) and M-CSF (CSF1), have been shown to be dispensable using various KO mouse models [5], FLT3L was identified to be a crucial factor for steady-state DC formation [28]. Via its receptor FLT3, it induces upregulation of the hematopoietic TF PU.1 and allows differentiation of HSPCs to all types of cDCs and pDCs. Accordingly, HSPCs lacking either *Flt3* or *Sfpi* (the gene encoding PU.1) showed defective DC differentiation potential both in vitro and upon in vivo transfer [8]. Interestingly, although PU.1 activation increases FLT3 expression, suggesting a positive feedback loop between Flt3 and PU.1, restoration of FLT3 expression in PU.1-deficient HSPCs did not restore their potential to give rise to DCs [8], indicating that in addition to FLT3, other targets of PU.1 are vital for DC differentiation. Here, we show that another myeloid TF, C/EBPα, is critical for steady state DC formation: HSPCs lacking *Cebpa* expression were markedly hampered to produce DCs after stimulation with FLT3L in vitro, and step-wise differentiation analysis revealed that C/EBPα was specifically needed for formation of MDPs and their transition to CDPs. This requirement of C/EBPα during early steps of DC formation became also obvious by analyzing DCs and their progenitors in a *Cebpa*^Cre^-EYFP reporter mouse model. We observed a gradual rise of *Cebpa*/EYFP^+^ cells from early (MPPs) to late (MDP/CDP) progenitors, but no further increase towards mature DCs. The fact that the majority but not all late progenitor and splenic DCs were positive for *Cebpa*/EYFP may be explained by an incomplete process of *Cre* recombination in a proportion of cells, as has been reported for this and other Cre-based reporter mouse models [15,23].

C/EBPα and PU.1 are well characterized to coordinate gene expression in HSPCs differentiation. In a recent paper, Pundhir and colleagues unraveled the kinetics by which PU.1 and C/EBPα coordinate enhancer and gene activity during early myeloid differentiation at the CMP level [29]. Through integrated analyses of enhancer dynamics, transcription factor binding, and proximal gene expression in *Cebpa* KO HSPCs, they identified a marked reduction in PU.1 binding at many enhancers active in early myeloid development, uncovering a surprising C/EBPα dependency for binding of PU.1. In accordance with our observed differentiation block at (or perhaps even before) the MDP stage, they demonstrated that loss of C/EBPα leads to a differentiation block preceding the granulocyte/monocyte progenitors (GMPs). Similar to MDPs during DC development, GMPs are a distinct precursor population during granulocytic/monocytic differentiation downstream of CMPs and MPPs. Since Pundhir and colleagues detected several examples of enhancers in which PU.1 binding is dependent on C/EBPα during early myeloid differentiation [29], it is tempting to speculate that C/EBPα acts as a pioneering factor on selected enhancers during early DC development, possibly via its switch/sucrose nonfermentable (SWI/SNF) domain. This SWI/SNF domain was shown to be important for enhancer binding in adipocyte differentiation, where C/EBPα is also known to be crucial [30]. Interestingly, overexpression of PU.1 was not able to restore DC formation in *Cebpa* KO HSPCs [16], suggesting that either its interaction with PU.1 or other PU.1-independent mechanisms induced by C/EBPα are crucial for DC development.

Since we observed downregulation of *Cx3cr1* in *Cebpa* KO HSPCs, and this could at least in part be overcome by addition of TNFα resulting in formation of fully functional, mature DCs, one could speculate that *Cx3cr1* is one such critical gene induced by C/EBPα. *Cx3cr1* expression was used as the earliest marker for identification of MDPs in murine bone marrow [31], and in silico analysis shows C/EBP binding sites in its promoter region (see Appendix A). Furthermore, expression of CX3CR1 protein promoted the generation of DCs under steady-state conditions [32]. Consistently, lack of CX3CR1 affected DC differentiation from bone marrow myeloid cells induced by CSF2 (GM-CSF) and interleukin-4 (IL-4) in vitro [33]. However, several other factors, including crucial TF during DC development, like IRF4, IRF8, KLF4, as well as C/EBPβ, were shown to be downregulated in *Cebpa* KO CMPs [16]. Thus, additional studies are clearly needed to unravel the definite molecular mechanisms regulated indispensably by C/EBPα during early DC development.

Through comparison of gene expression analysis in *Cebpa* WT and KO FLT3^+^ HSPCs, we have identified a compensatory role of TNFα in early DC development in the absence of C/EBPα. TNFα-induced genes were among the most prominently dysregulated genes in KO HSPCs and, even more intriguing, addition of TNFα to FLT3L could at least partially rescue mature DC formation in KO HSPCs. The role of TNFα in hematopoiesis has been extensively studied, but remains controversial. While TNFα was shown to affect DC development via NF-κB signaling [34] and also has been used for in vitro generation of DCs from human and murine HSPCs in combination with other cytokines [26], a recent study reported inflammation-induced elimination of differentiated myeloid progenitors by TNFα in the bone marrow [35]. In the same study, however, TNFα treatment of mice prevented necroptosis in hematopoietic stem cells and initiated emergency myelopoiesis through NF-kB-dependent mechanisms, hence promoting HSPC regeneration [35]. TNFα itself can directly and rapidly upregulate PU.1 protein in HSCs by recruiting NF-κB to bind to the enhancer element of PU.1 [36]. Concerning C/EBPα though, inflammatory cytokines were also shown to restore formation of GMPs and mature granulocytes from *Cebpa* KO fetal liver cells in culture and in vivo [37]. In mouse models of acute myeloid leukemia (AML) using *MLL-AF9* and *MOZ-TIF2* fusion genes, C/EBPα-induced formation of GMPs was critical for leukemia development, since deletion of *Cebpa* prevented initiation of AML. Interestingly, treatment of *Cebpa* KO HSPCs of these mice with inflammatory cytokines reestablished AML transformation capacity [38]. Together with our results, these data indicate that distinct inflammatory cytokines like TNFα can restore transcriptional activity in early myeloid differentiation in the absence of C/EBPα. However, the exact mechanisms involved in this process remain to be established.

## 5. Conclusions

Using the *Cebpa* reporter as well as inducible KO mouse models, we identified an important role of C/EBPα in early DC development. Specifically, C/EBPα was needed for the formation of MDPs and their transition to CDPs. The inflammatory cytokine TNFα at least partially rescued DC development in *Cebpa* KO HSPCs by overcoming this block in early differentiation.

## Figures and Tables

**Figure 1 cells-09-01223-f001:**
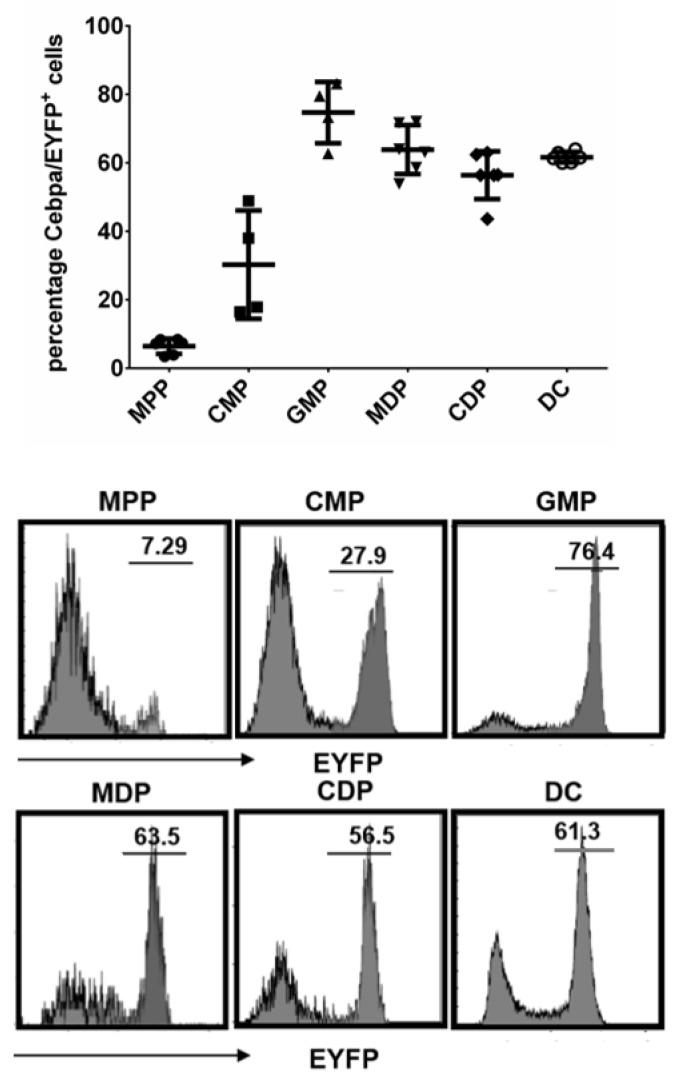
*Cebpa*/EYFP expression in distinct progenitor cell compartments and mature splenic CD11c^+^MHCII^+^ dendritic cells (DCs). Data represent the mean (+ SD) percentage of *Cebpa*/EYFP^−^ positive cells of the given subpopulation obtained from 4–6 mice.

**Figure 2 cells-09-01223-f002:**
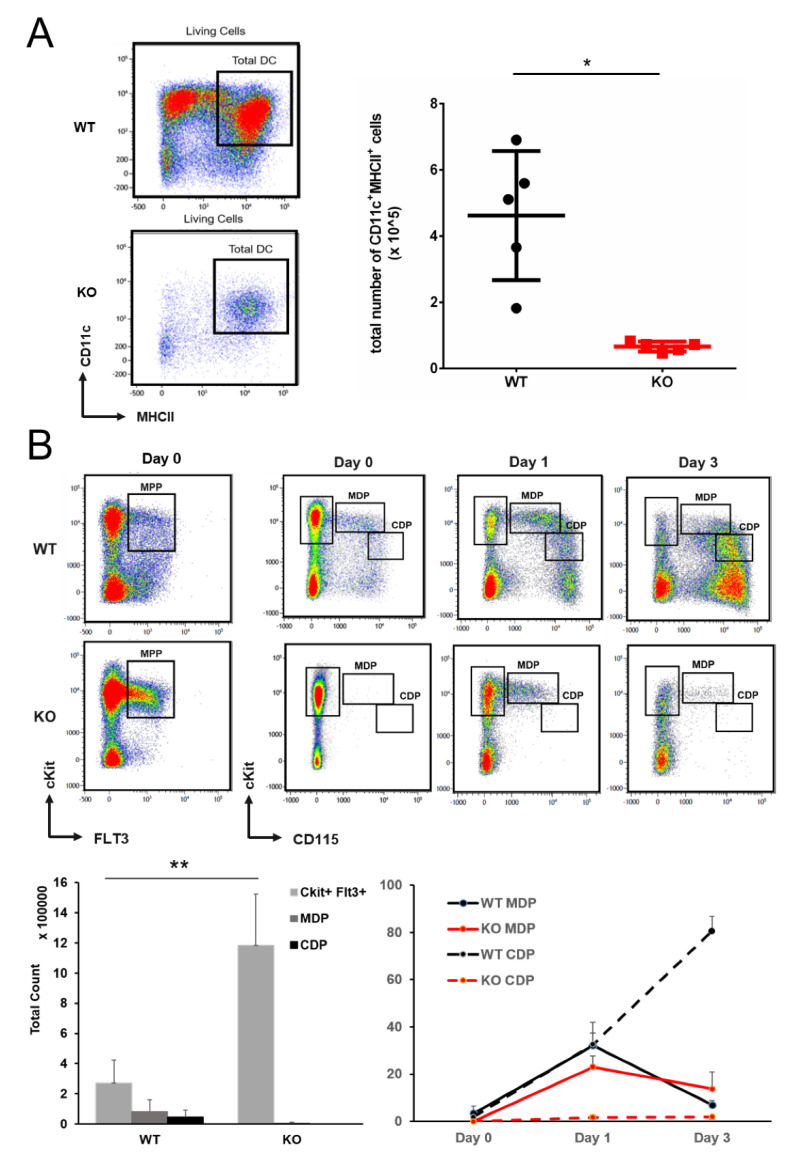
C/EBPα is needed for FMS-related tyrosine kinase-3 ligand (FLT3L)-induced DC development. (**A**) FLT3L-induced in vitro formation of mature CD11c^+^MHCII^+^ DC is significantly reduced in *Cebpa* KO, as compared to WT hematopoietic stem and progenitor cells (HSPCs) (*p* < 0.01, *n* = 5 mice per group). Results are representative from three independent experiments. A representative flow cytometry plot for the analysis of mature CD11c^+^MHCII^+^ DC is shown on the left side (upper panel: WT, lower panel: KO). (**B**) Despite higher numbers of FLT3^+^CD117^+^ progenitors at day 0 (lower left panel), analysis of early differentiation after FLT3L treatment in vitro reveals a reduced formation of monocytic dendritic cell progenitors (MDPs) and a block of transition towards common dendritic cell progenitors (CDPs) in KO HSPCs. Cells were analyzed by flow cytometry on days 0, 1, and 3 of culture (upper panels: WT, lower panels: KO). The percentage of MDPs and CDPs of total CD117^+^ HSPCs on days 1 and 3 are given in the right lower panel. All data represent mean + SD of 5 mice (* *p* < 0.01, ** *p* < 0.001).

**Figure 3 cells-09-01223-f003:**
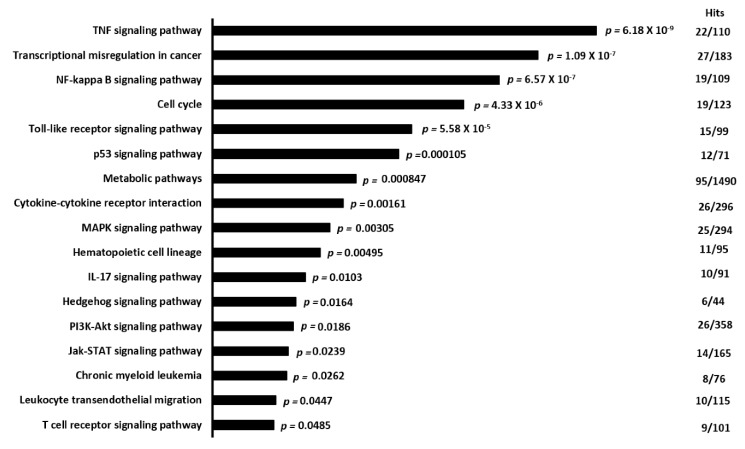
Pathway analysis of modulated genes between WT and KO HSPCs after FLT3L treatment. Bars represent the significance of hits. On the right, the number of differentially expressed genes (hits) as well as total number of genes per pathway are given.

**Figure 4 cells-09-01223-f004:**
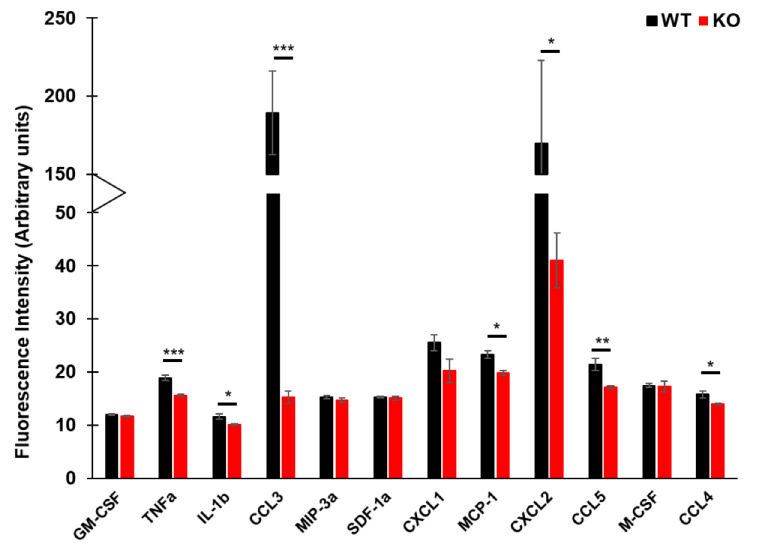
Bioplex analysis of various cytokines and chemokines from supernatants isolated from FLT3L-treated WT and KO HSPCs (*n* = 6 mice, * *p* < 0.01, ** *p* < 0.001, and *** *p* < 0.0001).

**Figure 5 cells-09-01223-f005:**
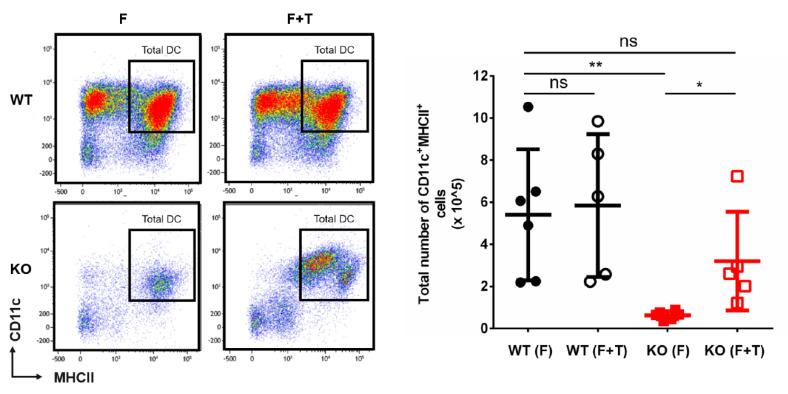
Addition of TNF rescues formation of mature DCs in the absence of *Cebpa*. In vitro formation of CD11c^+^MHCII^+^ DCs is increased, when KO HSPCs are cultured with a combination of FLT3L (F) and TNFα (T), as compared to FLT3L only. Data represent mean + SD (*n* = 5–6 mice, * *p* < 0.01, ** *p* < 0.001; NS denotes nonsignificant).

**Figure 6 cells-09-01223-f006:**
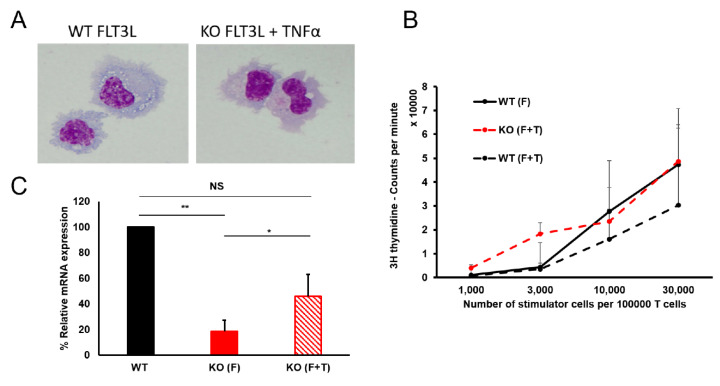
DCs formed from KO HSPCs by TNFα and FLT3L are functionally active. (**A**) Morphological comparison of DCs from WT HSPCs treated with FLT3L and DCs of KO HSPCs treated with TNFα and FLT3L for 8 days. (**B**) Allogenic MLR after 5 days of stimulation shows comparable T-cell activation potential of DCs from WT HSPCs treated with FLT3L and DCs of KO HSPCs treated with TNFα and FLT3L. Data represent mean + SD (*n* = 4–6 mice). (**C**) RT qPCR analysis shows an increase in expression of *Cx3cr1* in KO HSPCs treated for 4 h with TNFα and FLT3L, as compared to FLT3L only. Data represent fold change relative to WT normalized with two endogenous housekeeping genes (*n* = 3 mice, * *p* < 0.05, ** *p* < 0.01; NS denotes nonsignificant).

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
