# Peer review of "TNFα Rescues Dendritic Cell Development in Hematopoietic Stem and Progenitor Cells Lacking C/EBPα"

_cells, 2020, doi:10.3390/cells9051223_

Round 1

Reviewer 1 Report

In the manuscript titled “TNFa Rescues Dendritic Cell Development in Hemato-Poietic Stem and Progenitor Cells Lacking C/EBPa” by Anirudh et al., the authors demonstrate an important role for expression of the C/EBPa during FLT3L -dependent dendritic cell (DC) development. While a role for C/EBPa has previously been established, the work does provide some new insights into the process, particularly with respect to a possible connection between C/EBPa transcriptional regulation and TNFa -dependent developmental signals. The central novel finding of the work is that TNFa stimulation may be able to partially rescue the DC developmental defects associated with C/EBPa expression. While the topic is appropriate for the journal and the work is likely to be of interest to its readers, there are several limitations of the experimental design, lack of controls, and data interpretation that temper enthusiasm for the work in its present form. Specific concerns to address in revision are delineated below:

  • What is the authors’ interpretation of the 40% EYFP-negative splenic DCs (Figure 1)? It seems like this issue warrants at the very least a more comprehensive treatment in the Discussion section.
  • Related to point #1, I’m unsure how to reconcile Figure 2B with Figure 2A. In Figure 2B (Day 3), it looks like development of CDP is almost completely absent in the KO cells, so where are the CD11c/MHC-II -positive cells coming from in the KO in Figure 2A?
  • In Figures 4 and 5, the representation of the data in terms of “Fluorescence Intensity” as opposed to actual quantified amounts of cytokine makes it impossible to determine whether these levels and statistical differences might be actually biologically meaningful. Furthermore, the TNFa levels used for the rescue experiments later in the manuscript should correlate (in actual cytokine amounts) to the deficit experienced in the KO. It is hard to believe that the dose of TNFa used is physiologically relevant for homeostatic DC development in uninfected organisms. Also, the discussion could use some more in-depth discussion on the physiological relevance of TNFa -dependent DC differentiation/development.
  • Figure 5 is missing an important TNFa-treated WT control group.
  • Figure 6 is missing both the TNFa-treated WT control group and the FLT3L only KO control group. Without these controls, it is basically impossible to adequately interpret the results presented.
  • Although addressing this is not a “requirement” for the revision, MLRs are a pretty unsatisfying way to address T cell stimulatory capacity (particular in the mouse system with an abundance of more rigorous ways to test antigen-specific DC-mediated T cell activation).
  • At the heart of the story, it remains unclear whether the authors are arguing that the C/EBPa deficit functions specifically with respect to DC development (which argues what are the DCs that do develop! – see points #1 and #2); or, if they also want to claim that, in addition to the developmental deficit, there is a functional deficit in the mature DCs (which Figure 6B might argue if the relevant controls were in place), then TLR-mediated DC maturation and activation would be appropriate metrics to include as well.

Round 2

Reviewer 1 Report

The authors have meaningfully and effectively addressed the concerns brought forth in the original review.

Author Response

Thank you!

Author Response

Please see attachment, thank you!

Round 3

Reviewer 2 Report

Dear authors,

It was an honour to review the manuscript "TNFalpha rescues dendritic cell development in hematopoietic stem and progenitor cells lacking C/EBPα". Thank you very much for addressing all my raised comments and clarifying the last minor issues. 

As I wrote before, in my view, the presented manuscript is interesting, relevant and well performed. I have no further comments and would urge the editors to publish this elegant work.

Very best wishes